# An Intelligent Bat Algorithm for Web Service Selection with QoS Uncertainty

## Abdelhak Etchiali *, Fethallah Hadjila and Amina Bekkouche

Computer Science Department, University of Tlemcen, Tlemcen 13000, Algeria;
fethallah.hadjila@univ-tlemcen.dz (F.H.); amina.bekkouche@univ-tlemcen.dz (A.B.)
* Correspondence: abdelhak.etchiali@univ-tlemcen.dz

**Abstract:** Currently, the selection of web services with an uncertain quality of service (QoS) is gaining much attention in the service-oriented computing paradigm (SOC). In fact, searching for a service composition that fulfills a complex user's request is known to be NP-complete. The search time is mainly dependent on the number of requested tasks, the size of the available services, and the size of the QoS realizations (i.e., sample size). To handle this problem, we propose a two-stage approach that reduces the search space using heuristics for ranking the task services and a bat algorithm metaheuristic for selecting the final near-optimal compositions. The fitness used by the metaheuristic aims to fulfil all the global constraints of the user. The experimental study showed that the ranking heuristics, termed "fuzzy Pareto dominance" and "Zero-order stochastic dominance", are highly effective compared to the other heuristics and most of the existing state-of-the-art methods.

**Keywords:** web service selection; QoS uncertainty; bat algorithm; service-oriented computing





## 1. Introduction

With the advent of cloud computing and specifically online services (SaaS), it becomes more challenging to discover and select the best services with respect to the user's requirements [1,2]. Broadly speaking, we observe that a given functionality can be fulfilled by numerous SaaS with a variety of QoS levels. For complex user's requests (in terms of workflow), the task of selecting the best composition of services that satisfies the user's global constraints (e.g., the maximum cost of the composition of services is less than a given budget) is time consuming and far from meeting the user's expectations. It is worth noting that the selection of service compositions is NP-complete and exponentially depends on the number of tasks of the workflow. In general, the concepts related to the QoS, quality of experience (QoE), and end-to-end constraints are thoroughly defined in recommendation documents specified by the International Telecommunication Union (ITU) organization (see the recommendation identified by ITU-T Supp. 9 of the E.800 Series for more details about the regulatory facets of QoS). In practice, we observe that the QoS of SaaS applications is inherently uncertain and always changing; for instance, the cost of booking a hotel room or an airline ticket is uncertain and depends on the period (such as the season or month), social events, and other contextual aspects. To compare the services of the same functionality class while considering the different realizations of the QoS criteria, one can use statistical measures such as the mean QoS value or the median value to derive the best alternatives.

Unfortunately, these measures may not be effective and can yield misleading or unsatisfactory results. For instance, in the ensemble learning area, and more specifically the Adaboost method [3], it is known that the weighted average of different predictions (which is a special case of the mean value) can largely deviate from the true prediction if noise is present in the training set [4,5], and this means that the average value of a sample may not be the correct representative of a series if the noise is largely present in the data. In addition, according to the central limit theorem, the sample mean will approximately follow the normal distribution and will converge to the distribution expectation (for an

infinite size of the sample) if a given number of conditions are satisfied; otherwise, the sample mean will not be the best representative of the series (or the QoS attribute). These conditions involve, among others, the finiteness of the distribution variance, the sampling from the same distribution, the independence of the samples (which is hard to achieve in most scenarios), and the sufficiency of the number of samples. In summary, the sample mean is not the best representative of a series (excluding some exceptional cases).

In the same line of thought, we point out that the pertinence of service compositions with respect to the user's request is no longer a deterministic score, but it is rather specified as a probability of satisfying the global QoS constraints; this score is termed the global QoS conformance (GQC) [6]. As a result, the complexity of the selection issue is dependent on the number of tasks and also is impacted by both the size of each task and the size of the QoS sample (i.e., the number of realizations per QoS attribute).

The GQC can be also seen as the expected value of a random variable termed Z, where Z provides an outcome equal to 1 if the aggregated QoS satisfies the global constraint bound; moreover, it is highly desirable to obtain service compositions that satisfy a maximum number of global constraints in terms of the median QoS (this means that 50% of the solution realizations—of a single QoS attribute—will ensure the end-to-end bounds). This criterion is denoted as the percentage of satisfied global constraints (PSGC). This latter measure can be considered (sometimes) an alternative to the GQC objective function, since it ensures a high gain of computational cost.

For instance, according to Table 1 (where the global constraint (GC) is specified in the first line), we observe that only $(S_1, S_{10})$ and $(S_2, S_{10})$ will be retained as feasible solutions since the PSGC = 100% (the example comprises a single QoS attribute; in addition, the sum of 20 and 26 is greater than 45 for both pairs); however, the remaining compositions are not feasible, and therefore, the PSGC = 0 (in fact, the sum of the median QoS exceeds the global constraint bound).

**Table 1.** Motivating example.

| G.C: $AggregatedQoS(S_x, S_y) \geq 45$ | |
|:---:|:---:|
| **Task T1** | **Task T2** |
| $S_1:$ | $S_9:$ |
| $QoS(S_1) = < 5, 15, 20, 30, 70 >$ | $QoS(S_9) = < 10, 11, 15, 20, 22 >$ |
| $S_2:$ | $S_{10}:$ |
| $QoS(S_2) = < 6, 18, 20, 40, 90 >$ | $QoS(S_{10}) = < 15, 18, 26, 30, 40 >$ |
| $S_3:$ | $S_{13}:$ |
| $QoS(S_3) = < 4, 15, 18, 25, 250 >$ | $QoS(S_{13}) = < 3, 8, 10, 20, 30 >$ |

To summarize, our selection issue needs effective ranking heuristics for the workflow tasks, as well as time-efficient approaches for exploring the service compositions. To address these difficulties, we propose a two-stage approach that ensures high fitness service compositions and acceptable responsiveness delay.

In the first step, we reduce the search space in each task by only retaining the Top-K pertinent services in terms of a given heuristic $H_i$. Consequently, the total search space is reduced from $m^n$ candidate solutions to $k^n$ candidate solutions, where n stands for the number of tasks and m stands for the number of services per task (see Table 2 for more details).

In the second step, we perform a heuristic global search (this means that our solution will be a vector of n services) and retain the Top-K compositions in terms of the GQC.

Our contributions can be summarized as follows:

- We downsized our search space from $m^n$ to $k^n$ by retaining the most-pertinent elements of each task. To this end, we propose four ranking heuristics of the items of each task. All these heuristics perform pairwise comparisons of the services and select the Top-K elements having the maximum number of wins:
    - $H_1$ is an efficient implementation of the fuzzy Pareto dominance; it was inspired by [7].
    - $H_2$ (zero-order stochastic dominance) is a stochastic dominance relationship that uses the zero-order terms of the QoS sample [8]. It directly uses the QoS realizations during the comparisons.
    - $H_3$ (first-order stochastic dominance) is a stochastic dominance relationship that uses the first-order terms of the QoS sample [8]; this means that $H_3$ uses the cumulative distribution of the sample to perform comparisons.
    - $H_4$ (the majority interval heuristic) was inspired by [9]. In this ranking, we compute the median interval of each service and perform pairwise comparisons of the services using Equation (27). The services having the highest number of wins are retained in Top-K elements.
- In the second step, we performed a global search on the retained Top-K services using a swarm-intelligence-based algorithm termed the "discrete bat algorithm (DBA)". This metaheuristic was chosen because of its ability to leverage both global search operators and local search operators during the exploration of candidate solutions (in contrast to metaheuristics, which only use one operator, such as particle swarm optimization or ant colony optimization). The coordinated use of these operators can achieve promising results on NP-complete problems.
- At the end, we evaluated the effectiveness and efficiency of the approach using a consolidated set of experiments.

The rest of this paper is organized as follows: Section 2 presents a literature overview of the existing works. Section 3 specifies the problem statement. In Section 4, we introduce the proposed approach, as well as the selection algorithms. In Section 5, we present a set of experimental evaluations and compare our method with existing works. Section 6 concludes the paper and presents future perspectives.

## 2. State-of-the-Art

Selecting service compositions using QoS is a major topic in service-oriented computing (SOC). We mainly distinguish two categories: service selection with a certain (deterministic) QoS and service selection with an uncertain (nondeterministic) QoS. In what follows, we will review the two parts.

### 2.1. Service Selection with a Certain QoS

In this category, we assumed that the QoS attributes are static and do not change over time; therefore, the evaluation function of the compositions is also deterministic. Many works and reviews have been proposed to address this kind of issue [1,10–12]. In what follows, we will discuss the most-important ones.

The review presented in [13] specified two main categories for handling the QoS-aware service-selection problem: the exact algorithms and the approximate algorithms (heuristic/metaheuristic). In each category, the authors reviewed many tips and strategies to simplify the problem resolution, including the cost function linearization, the local QoS optimization, and the simple additive weighting. In [14], the authors proposed a framework that first takes the skyline services of each task; then, a set of service clusters (within each task) are hierarchically created using K-means to lower the size of the search space. At the end, the solutions are explored using the combinations of cluster-heads. The work by [15] decomposes global QoS constraints into local constraints using the culture genetic algorithm, then the top items are selected to aggregate the final compositions.

In [10], the service selection issue was viewed as an optimization problem that takes into account both functional (the function signature) and nonfunctional attributes (QoS,

global constraints) to select the Top-K service compositions. The objective function involves several parts, including a similarity function for the input/output matching, a utility function for assessing the aggregated QoS, and a penalty function for evaluating the satisfaction of global constraints. The authors leveraged the harmony search to derive the compositions that best meet the complex requirements. In [16], a multi-criteria decision method termed Topsis was proposed to handle the QoS-aware service selection. The overall idea consists of computing a distance between each candidate service and a couple of synthetic services termed the ideal positive item and the ideal negative item; the greater the distance is, the better the rank of the candidate element. The approach was tested on a small collection of six services, as well as three QoS attributes (cost, security, reliability). Despite the effectiveness of the results, the proposition needs scalable benchmarks to confirm its adequacy.

In [17], both local and global searches were leveraged for tackling the selection of cloud services. The proposed approach involves three steps: First, the REMBRANDT technique (which is a multi-criteria decision-making method) is applied to each task to select a subset of n services that have the best scores. Second, a pass of compatibility check is performed to further reduce the search space. Finally, a Dijkstra-based algorithm is applied to derive the optimal compositions in terms of the aggregated QoS and the number of cloud service providers.

The work by [18] tackled both the reliability assessment and the optimal selection of web service compositions. To estimate the reliability of complex web services, the authors adopted an extended version of PetriNet models and a mathematical model that leverages different factors including the network availability, the hermit device availability, the binding reliability, and the discovery reliability. To handle the second issue, a two-stage method was proposed: first the local skylines of each task of the workflow were extracted, then the global skylines were searched using R-tree structures and a multi-attribute decision-making method.

In [12], the authors viewed the web service-selection problem as an optimization of deterministic QoS attributes. More specifically, they designed an objective function that involves both an assessment of the aggregated QoS of service workflows and a penalty function for measuring the satisfaction degree of global constraints. In addition, a discretization of the continuous harmony search metaheuristic was proposed for performing the exploration of near-optimal compositions.

In [11], the authors used an optimized artificial bee colony (OABC) method for service composition. Mainly, the authors introduced three ideas into the initial bee algorithm: the first one is the diversification of the initial population; the second one is the dynamic adjustment of the neighborhood size of the local search; the third one is the addition of a global movement operator that aims to get closer to the global solution. The work by [19] leveraged fuzzy dominated scores to derive the Top-K services that have a more balanced QoS (and which can be better than some skyline services with undesirable QoS values) in a self-contained task. In [20], the authors considered the self-organizing migrating algorithm (SOMA) and the fuzzy dominance relationship to aggregate service workflows. The fuzzy dominance function was used in the SOMA metaheuristic to compute the QoS-aware distances between services. A bio-inspired method termed enhanced flying ant colony optimization (EFACO) was proposed in [21]. This approach constrains the flying activity and handles the execution time problem by a modified local selection. Since this phase may degrade the selection quality, a multi-pheromone approach was adopted to enhance the exploration through the pheromone assignment to each QoS criterion.

In [22], the authors clustered the cloud services using a trust-oriented k-means, then they created the composition of cloud services using honey bee mating. It is worth noting that the proposed framework is not scalable for large datasets. The work by [23] tackled the service-selection problem by handling multiple users' requirements. The approach is comprised of two steps: firstly, an approximate Pareto-optimal set is computed using

approximate dominance; secondly, the near-optimal compositions are selected using the artificial bee colony algorithm.

In [24], the authors proposed a hybrid recommendation method for predicting the missing QoS. The main idea consists of using both matrix factorization methods and the context of users and services to estimate the target QoS. The designed cost function involves a part from the latent factor model and a collaborative prediction model that uses the context-based neighbors. The results showed that the user context is more accurate than the service context, but the weighted average of both sub-models (the service context and the user context) is largely superior to the individual models.

In [25], the authors predicted the QoS of a web service (which can be involved in a composition) using linear regression and correlation checking. More specifically, the proposed approach uses two different QoS datasets: the first one contains nine quality levels (such as response time, availability, and reliability), and the second one comprises a set of source code metrics that cover the quantity metrics, complexity metrics, and quality metrics (a total of fifteen metrics). This collection of metrics is also known as Sneed's catalog. The objective of the study was to learn a multivariate linear model that predicts the level of a quality attribute from the variables of Sneed's catalog.

The work by [26] proposed a multi-stage composition method based on local and global optimization in addition to the handling of QoS flexibility. The proposition takes into account several types of workflows (including sequential, parallel, iterative, and conditional structures). The method first decomposes the global constraints into local constraints using well-defined heuristics; second, it relaxes the obtained bounds by adding/subtracting/multiplying flexibility terms and filters out the nonrelevant services. Third, the set of local Pareto-optimal services is extracted from each set of relevant services; finally, the Pareto-optimal compositions are computed using a progressive search. In [27], an automated planning algorithm called Graphplan was proposed to address the composition of land cover services. The key idea of the proposed framework consists of creating an ontology for describing the tasks, the input/output data, and the atomic services, then a planning graph is created using the forward search of the planning algorithm. This graph contains two types of layers, one for modeling the services and the second one for modeling the input/output data (also termed facts). The building of the service composition is performed during the backward search, which is guided with mutual exclusion constraints over both facts and services.

### 2.2. Service Selection with Uncertain QoS

The framework of [28] was one of the earliest works that addressed the service selection with uncertain QoS. The authors proposed an excellent heuristic, termed P-dominant skyline, to derive the best QoS-aware services in a self-contained task. The P-dominant skyline is considered to be resilient to QoS inconsistencies and noise. Moreover, this heuristic is accelerated using R-trees. A set of probability distributions was proposed in [29] to model the QoS uncertainty of service workflows. To select the best compositions, the authors used both integer programming and global constraint penalty cost functions.

A majority interval-based heuristic was introduced in [9] in order to derive the pertinent services of a set of tasks. The main idea consists of computing the median interval of each nondeterministic QoS attribute and comparing them using rectified linear unit (ReLU) functions [30]. After that, an exhaustive search is applied to obtain the final compositions. In [31], the authors proposed a set of heuristics for ranking the services of the workflow tasks. These propositions included probabilistic dominance relationships and fuzzy dominance alternatives. Once the Top-K elements are retained from each task, a constraint programming approach is applied to retain the Top-K optimal compositions of the services. In [32], the authors addressed the service composition issue by handling the QoS uncertainty and the location awareness. They proposed a sophisticated approach that combines the Firefly metaheuristic with a fuzzy-logic-based web service aggregation.

The framework proposed in [33] sorted the services of each task using both the entropy and the variance of the QoS attributes. The services that have larger values in terms of entropy and variance are discarded since they are considered as noisy or inconsistent services. Then, the items that have the lowest entropy/variance scores were retained to compose the final solutions.

The framework proposed in [6] was one of the first works that handled the QoS uncertainty and composition at the same time. Based on ideas defined in [34], the strategy adopted by the authors consisted of decomposing the end-to-end constraints into local constraints; the local edges (entrances) are calculated by dividing the end-to-end constraint bounds in proportion to the aggregated median QoS of each class of the workflow. After that, an initial service composition is built using a predefined utility function. If this latter one is not optimal, the method searches for alternative solutions using simulated annealing. In the same line of thought, the authors in [35] introduced a proposition for web service selection with the presence of outliers. Contrary to the work of [6], this method leverages a different heuristic to divide the end-to-end constraints into local constraints. The proposed idea ensures a high resilience against outliers (services with a noisy or unusual QoS). The work by [36] leveraged the stochastic dominance relationship to sort the services of each task of the user's workflow; after that, a backtracking search is applied to the filtered tasks to derive optimal service compositions. In [37], the authors proposed an interval-based multi-objective bee colony method to address the uncertain QoS-aware service-composition problem. The authors proposed an interval-oriented dominance relationship for comparing the services using intervals that represent the variation range of the QoS attributes. In addition, an interval-valued utility function was introduced to assess the quality of a composition with QoS uncertainty. Finally, an improved version of NSGA-II was used to derive the non-dominated service compositions. The framework proposed in [38] involved two steps: the first one retains the pertinent services of the local tasks using majority grades, and the second step performs a constraint programming search to keep the optimal compositions. In the same line of thought, the work by [39] proposed a heuristic for filtering the desirable services of each local task using hesitant fuzzy sets and cross-entropy, then a metaheuristic termed grey wolf optimization was applied to retain the Top-K near-optimal service compositions. In [40], the authors proposed a framework based on intuitionistic fuzzy logic to model the uncertainty of service compositions. It is worth noting that intuitionistic fuzzy logic is an extension of fuzzy logic in which the imprecise sets are modeled using three quantities: the membership degree, the non-membership degree, and the uncertainty degree. The authors targeted both single service devices and a type of device composition (with a parallel structure). Regarding device compositions, the authors proposed two mathematical models for estimating the uncertainty of data traffic quality. The first one uses the intuitionistic fuzzy information and internal parameters of the service components, while the second one uses only the intuitionistic fuzzy values of the component devices.

In what follows, Table 2 summarizes the most-important properties of some prominent approaches; in addition, the abbreviation "nop" means near-optimal.

**Table 2.** Recapitulation of most-prominent methods.

| Criterion / Method | [38] | [39] | [31] | [35] | [6] | [16] | [20] | [17] | [18] | [10] | [12] |
|---|---|---|---|---|---|---|---|---|---|---|---|
| QoS Uncertainty | yes | yes | yes | yes | yes | no | no | no | no | no | no |
| Local Search | yes | yes | yes | yes | yes | yes | yes | yes | yes | no | no |
| Global Search | yes | yes | yes | no | no | no | yes | yes | yes | yes | yes |
| Global Constraints | yes | yes | yes | yes | yes | no | yes | yes | yes | no | no |
| Optimality | yes | nop | yes | no | no | yes | no | yes | yes | nop | nop |
| Handling Semantics | no | no | no | no | no | no | no | no | no | yes | no |
| Use of metaheuristics | no | yes | no | no | yes | no | yes | no | no | yes | yes |

## 3. Problem Specification

In what follows, we introduce the formalism used in handling the selection of service compositions with QoS uncertainty.

### 3.1. Parameter Notation

To tackle our problem, we used the notation shown in Table 3. We assumed that the user's workflow is composed of $n$ sequential tasks $cl_1$, $cl_2$, ..., $cl_n$, and each task is achieved by a service $s_i$ that has $r$ QoS attributes. Each QoS criterion is materialized by a sample of $l$ realizations (see Table 3 and Figure 1).

**Table 3.** Notations.

| Parameter | Semantics |
|---|---|
| $n$ | The number of tasks (classes). |
| $m$ | The number of services per task. |
| $r$ | The number of QoS criteria. |
| $l$ | The number of QoS realizations (i.e., the sample size). |
| $cl_1$, $cl_2$, ..., $cl_n$ | The set of tasks; each task involves atomic SaaS with the same functionality and a different QoS. |
| $s_1$ (respectively $s_2$, ..., $s_m$) | Represents the id of the selected service related to $cl_1$ (respectively $cl_2$, ..., $cl_n$). |
| $QoS_{piju}$ | The value of the pth QoS attribute related to the uth instance of the service $S_i \in cl_j$. |
| $b_1$, $b_2$, ..., $b_r$ | The user's global constraints (i.e., the bounds that need to be satisfied by the QoS of the composition). |
| $w_1$, ..., $w_r$ | The weight of the QoS attributes; the default value of each $w_p$ is $\frac{1}{r}$. |
| $k$ | The size of the outcome list (of compositions). |

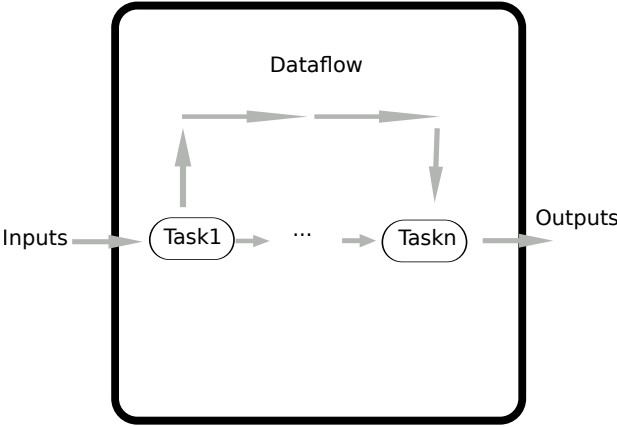

**Figure 1.** A general sequential workflow.

*3.2. QoS Model*

In this work, we only considered positive QoS attributes (i.e., those that need to be maximized). For negative attributes, we simply multiplied them by $-1$ and treated the new versions as positive ones. We note that our workflow is composed of n sequential tasks. The aggregated QoS of a workflow (having different patterns such as sequence, loops, parallelism, and choice) was presented in [14,41].

*3.3. Global QoS Conformance*

The measure of the global QoS conformance (GQC) [6] was leveraged to rank the Top-K compositions. The GQC is the probability that the composition of services satisfies all global constraints (see Equation (1)). In particular, we say that a composition $C$ is better than another composition $C'$ if the GQC of $C$ is higher than that of $C'$ with respect to Equation (1). If $C$ ties with $C'$, then we sort them according to the utility function ($U(.)$) that is shown in Equation (4); the larger the score of $U(.)$, the better the rank is.

Our aim was to search the compositions $C(s_{w1}, \ldots, s_{wn})$ such that the GQC is maximized:

$$
\begin{aligned}
GQC((S_{w_1}, \cdots, S_{w_n}), (b_1, \cdots, b_r)) = \\
\prod_{p=1}^{r} CC((S_{w_1}, \cdots, S_{w_n}), b_p)
\end{aligned}
\tag{1}
$$

Since we assumed that the QoS criteria are independent, the global QoS conformance is defined as the product of the constraint conformance (CC for short).

The criterion CC is defined as:

$$
\begin{aligned}
CC((S_{w_1}, \cdots, S_{w_n}), b_p) = \\
\frac{1}{l^n} \sum_{u_1=1}^{l} \cdots \sum_{u_n=1}^{l} step(aggregate(QoS_{pw_1u_1}, \cdots, QoS_{pw_nu_n}), b_p)
\end{aligned}
\tag{2}
$$

The function CC computes the satisfaction degree of a single global constraint. Finally, the binary function "Step" is defined as:

$$
Step(Aggregate(QoS_{pw_1u_1}, \cdots, QoS_{pw_nu_n}), b_p) = \\
\begin{cases} 1 & \text{if } Aggregate(s_{pw_1u_1}, \cdots, QoS_{pw_nu_n}) \geq b_p \\ 0 & \text{otherwise} \end{cases}
\tag{3}
$$

$$
U(C) = \sum_{p=1}^{r} w_p * \frac{(MedianQ'_p(C) - Qmin'(p))}{(Qmax'(p) - Qmin'(p))}
\tag{4}
$$

$$
Qmin'(p) = \sum_{j=1}^{n} Qmin(j, p)
\tag{5}
$$

$Qmin'(p)$ is the minimal aggregated QoS of the $p$th attribute for all possible compositions.

$$
Qmax'(p) = \sum_{j=1}^{n} Qmax(j, p)
\tag{6}
$$

$Qmax'(p)$ is the maximal aggregated QoS of the $p$th attribute for all possible compositions.

Equations $Qmin(j, p), Qmax(j, p)$ are defined as follows:

$$
Qmin(j, p) = Min_{u \in \{1, \ldots, l\}, s_i \in cl_j}(QoS_{piju})
\tag{7}
$$

$Qmin(j, p)$ is the minimal QoS value of the $p$th attribute of all services related to the $i$th task.

$$Qmax(j, p) = Max_{u \in \{1,\ldots,l\}, s_i \in cl_j}(QoS_{piju}) \tag{8}$$

$Qmax(j, p)$ is the maximal QoS value of the $p$th attribute of all services related to the $i$th task.

$$MedianQ'_p(C) = \sum_{j=1}^{n} Median_{u \in \{1,\ldots,l\}} QoS_{ps_j ju} \tag{9}$$

By assuming that the criterion $p$ is positive, the global constraint with respect to the median value is specified as:

$$MedianQ'_p(C) \geq b_p; \forall p \in \{1, \ldots, l\} \tag{10}$$

By assuming that the $p$th attribute is aggregated with a sum function, $MedianQ'_p(C)$ represents the aggregated QoS of $C$ with respect to the median QoS value of each component of $C$ (of the $p$th attribute).

Equation (10) is used to determine whether the global constraints are respected or not by the composition $C$. The PSGC is the ratio of constraints (in the form of Equation (10)) that are satisfied by a given composition.

To clarify the computation of the previous equations, we continue with the example cited in Table 1:

- $MedianQ'_p(C = <s_1, s_{10}>) = 20 + 26 = 46 \geq 45$.
- $GQC(C) = 16/25 = 0.64$. If $Qmin(1,1) = Qmin(2,1) = 0$ and $Qmax(1,1) = Qmax(2,1) = 300$, then:
- $U(C) = \frac{46-0}{(600-0)} = 0.075$. The composition $C$ is feasible. However, if the components of $C'$ are $<s_3, s_{13}>$, then:
- $MedianQ'_p(C' = <s3, s_{13}>) = 18 + 10 = 28 \leq 45$.
- $GQC(C') = 9/25 = 0.36$.
- $U(C') = \frac{28-0}{(600-0)} = 0.046$.

The composition $C'$ is not feasible because it violates the global constraint.

## 4. Proposed Approach

In what follows, we present the architecture of the proposed solution, as well as the different implemented algorithms (see Figure 2).

### 4.1. Overall Architecture

Our proposed framework involves three principal parts:

- The workflow building and update module: Its goal is to assign the new services to their corresponding tasks (a task is a functionality available on the Internet, e.g., hotel booking). This component also updates the tasks by changing/removing the services.
- The QoS update and management module: It stores all the QoS realizations of all services in a data warehouse; the QoS information may stem from different sources such as social networks (e.g., ratings, fidelity), third parties (e.g., throughput, latency), and service providers (e.g., cost).
- The QoS-aware service-selection engine: Given a user's workflow and the set of global constraints, the selection module allows searching the Top-K pertinent service compositions. As mentioned in the sequel, this engine achieves two steps: a local optimization (or sorting) and a global optimization. The first phase (local optimization) uses a set of heuristics (see Equations (15), (20), (24), and (28)) to rank the services of each task. The primarygoal is to downsize the search space by only keeping the first k services in the next phases.

The second phase of the engine performs a global optimization on the previous results. This step is realized using a discrete bat algorithm.

## Selection Framework

**Figure 2.** Service-selection architecture.

### 4.2. Local Optimization

In the following, we introduce four heuristics $(H_1, H_2, H_3, \text{and } H_4)$ that retain a subset (of size k) of each task. These services are the most-promising items in terms of each $H_i$. In this work, we assumed that the higher the value of a QoS level, the better the service.

#### 4.2.1. Fuzzy Pareto Dominance Heuristic (H1)

Many alternatives are available for implementing the fuzzy version of Pareto dominance [7,31,42,43]. To compare 02 r-dimensional vectors $u_d$ and $v_d$, we used the implementation specified in [7] since it is slightly more effective than the remaining alternatives and has zero hyper-parameters (in contrast to the others). Its definition is given in (12). The elementary fuzzy dominance (EFD) compares two scalar QoS values using Equation (11).

$$EFD(u_d(j), v_d(j)) = \begin{cases} 1 & \text{if } u_d(j) \geq v_d(j) \\ \frac{MIN(u_d(j), v_d(j))}{v_d(j)} & \text{otherwise} \end{cases} \tag{11}$$

$$FD(u_d, v_d) = \prod_{i=1}^{l} EFD(u_d(i), v_d(i)) \tag{12}$$

We assumed that $u_d$ and $v_d$ represent the values of the *d*th QoS attribute of two existing services S and S′ (respectively). To compare S and S′ with respect to all QoS attributes, we use Equation (13) (aggregated fuzzy dominance (AFD)).

$$AFD(u, v) = \prod_{d=1}^{r} EFD(u_d, v_d) \tag{13}$$

The fuzzy contest (FC) function shown in Equation (14) inspects the fuzzy dominance power of a service w with respect to another service q.

$$FC(S_w, S_q) = \begin{cases} 1 & \text{if } AFD(S_w, S_q) \geq AFD(S_q, S_w) \\ 0 & \text{otherwise} \end{cases} \tag{14}$$

Equation (15) computes the sorting score of a service $S_w$ by achieving a comparison with the rest of the candidate services of the current task (the larger the score, the better the rank).

$$FD\_SCORE(S_w) = \frac{1}{m-1} \sum_{w \neq q} FC(S_w, S_q) \tag{15}$$

In the experimental study, we sorted the services of each task according to the decreasing order of Equation (15) and took the first k elements.

We illustrate the principle of H1 by comparing the services $S_1$ and $S_2$ of Table 1:

If we apply Equation (13), we obtain

$AFD(QoS(S_1), QoS(S_2)) = FD(QoS(S_1), QoS(S_2)) = \frac{5}{6} \times \frac{15}{18} \times \frac{30}{40} \times \frac{70}{90} = 0.40$.

On the other hand:

$AFD(QoS(S_2), QoS(S_1)) = 1$ Consequently, $FC(S_1, S_2) = 0$, $FC(S_2, S_1) = 1$, $FC(S_2, S_3) = 1$, $FD\_SCORE(S_2) = 1$.

### 4.2.2. Zero-Order Stochastic Dominance (H2)

This heuristic compares the services using the raw QoS values [8] (see Equation (16)).

$$ZSD(u_d, v_d) = \frac{1}{l} \sum_{i=1}^{l} Step(u_d(i), v_d(i)) \tag{16}$$

$$Step(u_d(i), v_d(i)) = \begin{cases} 1 & \text{if } u_d(i) \geq v_d(i) \\ 0 & \text{otherwise} \end{cases} \tag{17}$$

To compare two S and S′ with respect to all QoS attributes, we use Equation (18) (aggregated zero-order stochastic dominance (AZSD)).

$$AZSD(u, v) = \prod_{d=1}^{r} ZSD(u_d, v_d) \tag{18}$$

To perform the majority vote (within a task), we need to compare each pair of services. To do so, we leveraged the contest function shown in Equation (19); it is termed the aggregated zero-order stochastic dominance contest (AZSDC) The AZSDC returns 1 if $S_w$ dominates $S_q$ (in the sense of AZSD); otherwise, it returns 0.

$$AZSDC(S_w, S_q) = \begin{cases} 1 & \text{if } AZSD(S_w, S_q) \geq AZSD(S_q, S_w) \\ 0 & \text{otherwise} \end{cases} \tag{19}$$

Equation (20) calculates the sorting score of a service $S_w$ by achieving a comparison with the rest of the candidate services of a given task (the larger the score, the better the rank).

$$ZSD\_SCORE(S_w) = \frac{1}{m-1} \sum_{w \neq q} AZSDC(S_w, S_q) \tag{20}$$

In the experiments, we sorted the services of each task according to the decreasing order of Equation (20) and took the first k elements.

### 4.2.3. First-Order Stochastic Dominance (H3)

Like H2, the first-order stochastic dominance (H3) [8] performs the same steps, except that it processes the cumulative distribution (CumulDistr) of the sample instead of the raw QoS. This heuristic is specified in Equation (21). If we assume that $u_d$ is the QoS sample of

the $d$th attribute of a given service $S$, then the cumulative distribution of $u_d$ is approximated as follows:

$u_d'(i) = CumulDistr_i(u_d) = \sum_{t=1}^{i} \frac{1}{l}$.

In addition, we increased the resolution (size) of $u_d'$ and set it to $2 \times l$; the added entries (i') will have a score equal to $\frac{u_d'(i-1)+u_d'(i)}{2}, \wedge i' \in [i-1, i]$.

$$FSD(u_d', v_d') = ZSD(CumulDistr(u_d), CumulDistr(v_d))$$

$$= \frac{1}{2 \times l} \sum_{i=1}^{2 \times l} Step(u_d'(i), v_d'(i)) \tag{21}$$

We have the same expressions mentioned in H2 for the rest of the equations.

$$AFSD(u', v') = \prod_{d=1}^{r} FSD(u_d', v_d') \tag{22}$$

$$AFSDC(S_w, S_q) = \begin{cases} 1 & \text{if } AFSD(S_w, S_q) \geq AFSD(S_q, S_w) \\ 0 & \text{otherwise} \end{cases} \tag{23}$$

$$FSD\_SCORE(S_w) = \frac{1}{m-1} \sum_{w \neq q} AFSDC(S_w, S_q) \tag{24}$$

In the experiments, we sorted the services of each task according to the decreasing order of Equation (24) and took the first k elements.

### 4.2.4. Majority Interval Dominance (H4)

In this heuristic, we first computed the median interval for each QoS attribute of each service (this means that the $d$of each service $S_x$ is represented by an interval $[lb_{x,d}, ub_{x,d}]$). Then, we ranked the services by comparing these representative intervals. To elucidate this idea, we considered the services S1 and S2 of Table 1. The median interval of S1 is $[5, 30]$, and the corresponding one of S2 is $[18, 40]$. To compare the median intervals, we used the function presented in [9]; this function is defined in Equation (25), and it is termed majority interval dominance (MID). (We assumed that the compared services $S_x$ and $S_y$ belong to the task j, and the current QoS attribute is d; $S_x$ is represented by $[a_1, a_2]$, and $S_y$ is represented by $[b_1, b_2]$.)

$$MID([a_1, a_2], [b_1, b_2]) = \frac{ReLU(a_1 - b_1) + ReLU(a_2 - b_2)}{2 \times (Qmax(j, d) - Qmin(j, d))} \tag{25}$$

where ReLU [30] is the activation function used in deep learning.

For instance, if we assume that $Qmax(j, d) = 300, Qmin(j, d) = 0$, then $MID(S_1, S_2) = MID([5, 30], [18, 40]) = 0$ and $MID(S_2, S_1) = MID([18, 40], [5, 30]) = \frac{13+10}{2 \times 300} = 0.038$.

The aggregated majority interval dominance is shown in Equation (26).

$$AMID(u, v) = \prod_{d=1}^{r} MID(u_d, v_d) \tag{26}$$

$u_d, v_d$ represent the median intervals of the compared QoS attributes (having the $d$th rank). Like H1, H2, and H3, the contest function is defined in Equation (27):

$$AMIDC(S_w, S_q) = \begin{cases} 1 & \text{if } AMID(S_w, S_q) \geq AMID(S_q, S_w) \\ 0 & \text{otherwise} \end{cases} \tag{27}$$

$$MID\_SCORE(S_w) = \frac{1}{m-1} \sum_{w \neq q} AMIDC(S_w, S_q) \tag{28}$$

In the experiments, we sorted the services of each task according to the decreasing order of Equation (28) and took the first k elements (Algorithm 1).

---

**Algorithm 1:** Discrete bat algorithm (DBA).

---

**Input:**
$< TopKList_1, \ldots, TopKList_n >$: the input lists given by the local optimization heuristics.
GC: the global constraints' bounds.
k: the size of the result list
**Output:**
TopKCompositions: the Top-K compositions that best meet tall global constraints in terms of the GQC (it is initially empty).

**1** $A \leftarrow ones(PopSize)$
  $R \leftarrow random(PopSize)$
  $Alpha \leftarrow 0.8$
  $Gamma \leftarrow 0.8$
  **for** $i \leftarrow 1$ *to PopSize* **do**
    $Bat_i \leftarrow RandomPosition(TopKList_1, \ldots, TopKList_n)$
    $Freq_i \leftarrow random()$
**2** **end**
**3** $Bat^* \leftarrow ArgMax_{i \in \{1,\ldots,PopSize\}}(GQC(Bat_i))$

**4** **while** $(it \leq MaxIt)$ **do**
**5**   **for** $i \leftarrow 1$ *to PopSize* **do**
**6**     $Bat_i \leftarrow GlobalMovement(Bat_i, Freq_i, Bat^*)$
      $Freq_i \leftarrow random()$
      **if** $(random() \geq R_i)$ **then**
**7**         $neighborhoodSize \leftarrow round(k * mean_{i \in \{1,\ldots,PopSize\}}(A_i)$
        $NewPosition \leftarrow neighbor(Bat^*, neighborhoodSize)$
**8**     **end**
**9**     **if** $(random() \leq A_i$ and $GQC(NewPosition) \geq GQC(Bat_i))$ **then**
**10**       $Bat_i \leftarrow NewPosition$
      $A_i \leftarrow Alpha \times A_i$ /*decrease the loudness rate*/ ;
**11**       $R_i \leftarrow 0.1 \times (1 - exp(-Gamma \times it)$ /*increase the pulse emission rate*/ ;
**12**     **end**
**13**     **if** $GQC(Bat_i) \geq GQC(Bat^*)$ **then**
**14**       $Bat^* \leftarrow Bat_i$;
**15**     **end**
**16**   **end**
**17**   $it \leftarrow it + 1$
**18** **end**
**19** $TopKCompositions \leftarrow update(TopKCompositions, \{Bat^*, Bat_1, Bat_2, \ldots, Bat_n\})$
  **return** $TopKCompositions$

---

### 4.3. Global Optimization

Once the n lists are given by the first step of the method, it is now time to perform a global search by composing and assessing the service compositions. To do so, we leveraged a swarm-intelligence-based metaheuristic that adapts the bat algorithm to our discrete context. This discrete optimization algorithm was chosen because of its ability to combine local search and global search in a harmonious way. The bat algorithm [44] is a promising metaheuristic for continuous optimization. Its metaphor is based on the echolocation behavior of micro-bats, which can vary the frequencies, loudness, and pulse-emission rates to capture prey (see Figure 3).

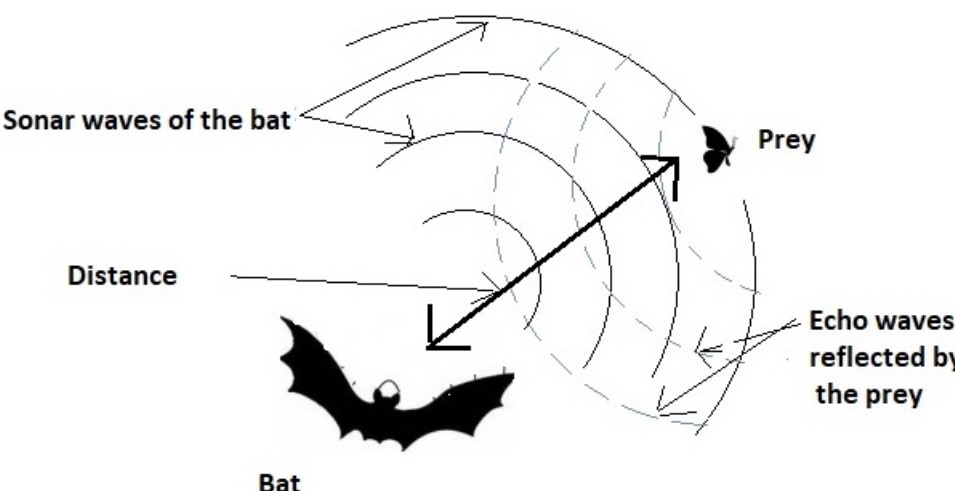

**Figure 3.** Bat metaphor.

Before detailing the pseudo-code of the discrete bat algorithm (see Algorithm 1), we explain all its technical parameters:

- Pop: This is a matrix of PopSize*n dimensions; it represents all the virtual bats. $Pop = \{Bat_1, \ldots, Bat_{PopSize}\}$.
- $Bat^*$: The position of the best bat.
- A: This stands for the loudness of the chirp; it is a vector of Popsize random numbers in [0, 1]l it controls the neighborhood size of the local search. It is decreased along the execution of the metaheuristic.
- Freq: This stands for the frequencies of the bats. It is a matrix of PopSize*n dimensions; it controls the size of the moving step during the global search phase. It is initialized with random values between 0 and 1.
- R: This stands for the pulse-emission rate of each bat. Technically, it is an n-dimensional vector of random numbers (in [0, 1]) that controls the execution of the local search.
- Alpha: The decreasing factor of A.
- Gamma: This is a factor that controls the increasing of the pulse-emission rate R.
- MaxIt: The maximum number of iterations of the DBA.

The pseudo-code of the DBA can be explained as follows:

- Line 1: For each bat, we initialized its loudness and pulse-emission rate. Furthermore the updating rates Alpha and Gamma were initialized.
- Line 2: For each bat, we randomly initialized its position and its frequency, which is used as a step displacement in the GlobalMovement (of Line 6). $Freq_i$ is a real value belonging to [0, 1].
- Line 3: We computed the best bat position of the swarm in terms of the GQC. We updated the best bat position of the swarm.
- Lines 4–18: This is the principal loop of the metaheuristic; it is constituted of the MaxIt.
- Lines 5–16: This is the loop that explores all the bats.
- Line 6: This function creates a new composition by moving toward the best solution with a random step. More specifically, for each component (task) j of a given bat i, we replaced it with the corresponding value in $bat^*$ with a probability equal to $Freq_i(j)$ ($Bat_i(j) = Bat^*(j)$, with a probability$= Freq_i(j)$). The frequency of each bat is changed after that.
- Lines 7–8: with a probability $1 - R_i$, we created a neighborhood centered on the best bat $Bat^*$.
- The width of this neighborhood is equal to k times the average of all the possible loudness $A_i$ (this width is termed spread); then, we created a new composition NewPosition $= (component_1, \ldots, component_n)$ as follows:

For each $j \in \{1, \ldots, n\}$ $component_j = successor_{Task_j}(Bat^*(j))$ with a probability $= Gaussian_{mean,\sigma}(|Rank(Bat^*(j)) - Rank(successor_{Task_j}(Bat^*(j)))|)$, knowing that $Mean = Rank(Bat^*(j))$ and $\sigma = spread/2$.

- For instance, if a task j is constituted of the following ranked services $< S9, S15, S4, S20, S2 >$, and we assumed that $Bat^*(j) = S4, mean = Rank(Bat^*(j)) = 3$ (it is ranked third in the list), and $\sigma = spread/2 = 1$, then the neighborhood of S4, according to Line 7, is equal to $\{S15, S4, S20\}$. The probability of obtaining each of them as a value for $component_j$ is 25%, 50%, 25%, respectively (since we approximated the Gaussian function for these three observations).
- In Line 9, we accepted the aforementioned solution NewPosition (i.e., we updated $Bat_i$), with a probability $A_i$. In addition, NewPosition must have a fitness better than that of $Bat_i$.
- In Line 10, we decreased the loudness $A_i$.
- In Line 11, we increased the pulse-emission rate $R_i$ to reduce the chances of performing the local search in the future (i.e., Line 7).
- In Lines 13–15, we updated the best solution if the actual bat had a better fitness.

Finally, we note that DBA has a time complexity of $O(PopSize \times (1 + n + r \times l^n + Maxit \times (n + n \times k + r \times l^n))$. We note that the complexity of the fitness function GQC is $O(r \times l^n)$.

## 5. Experimental Study

Inspired by [6,35], we generated the QoS dataset using a random Gaussian distribution. In particular, we used the following setting: mean = 0 and standard deviation = 1. The domain of each parameter is given in Table 4.

**Table 4.** Parameters' range.

| Parameter | Meaning | Domain | Default Value |
|---|---|---|---|
| n | The number of tasks | {2, 5, 8} | 2 |
| m | The number of services per class | {500, ..., 1200} | 500 |
| r | The number of QoS attributes | {4, ..., 11} | 4 |
| l | The number of realizations of a given QoS attribute (i.e., the number of instances) | {15, ..., 350} | 21 |
| k | The size of the returned list | {2, 5, 10} | 5 |
| $b_i$ | The $i$th global constraint bound | Positive real | For attributes aggregated with: an additive function: $n \times 0.6$. a multiplicative function: $0.6^n$. MAX/MIN functions: 0.6. |
| $w_i$ | The weight of the $i$th QoS attribute | [0, 1] | 1/r |

The experiments were implemented using a Windows10 64-bit OS with an Intel Core i3-6006U CPU @ 2.0 GHz processor and 32 GB of RAM. The algorithms were developed with Netbeans IDE 12.0.

Before introducing the experimental results, we describe the theoretical complexity of the proposed heuristics. The heuristic H1 (Equation (15)) compares each candidate service with the remaining components, and each comparison step (Equation (13)) is $O(r.l)$; therefore, the time complexity of H1 is $O(m.r.l)$. Like H1, the complexity of H2 (Equation (20)) is $O(m.r.l)$. In the same line of thought, the complexity of Equation (24) (H3) is $O(m.r.l)$, and the complexity of Equation (28) (H4) is $O(l.logl + m.r)$.

In the experiments, we only varied one parameter and kept the remaining set to their default values (see Table 4). As regards the fuzzy dominance implementation of [31], we preserved the same setting chosen by the authors for the parameter $\varepsilon$ (which is equal to 0.1). For the sake of concise presentation, we only show the Top-2 pertinent compositions (in terms of the GQC) for all the remaining experiments.

As shown in Figure 4, we observed that the behaviors (time) of H1, H3, and the heuristic of (21) were comparable. Additionally, we observed a slight rise of time for H3 since the curve slope is proportional to $2 \times l$ instead of l. In the end, we note that H2 and H4 were the most-efficient heuristics since the slope of their curves was lower than that of the first ones.

Figure 5 shows that the CPU times of H1, H3, and the heuristic of [31] are comparable, but the slopes of their respective curves were different. Additionally, we observed a slight rise of time for the heuristic of [31] since its complexity is quadratic with respect to l. We note that H4 was the most-efficient one since the comparison of median intervals does not depend on l (we assumed that the sorting of QoS vectors is performed in an offline way).

Like the previous experiments, Figure 6 shows that the fuzzy dominance implementations of (21), H1, and H3 had closer CPU times. On the other hand, the majority grade heuristic (28) and H4 had a lower CPU time since their theoretical slope is not dependent on l. We also note that the curve of H2 had an almost flat slope, and this was mainly due to the low overhead of Equation (16). It is worth noting that the majority grade principle was initially presented by [45] for ranking the candidates of an election. After that, it was adapted by [38] to web service selection.

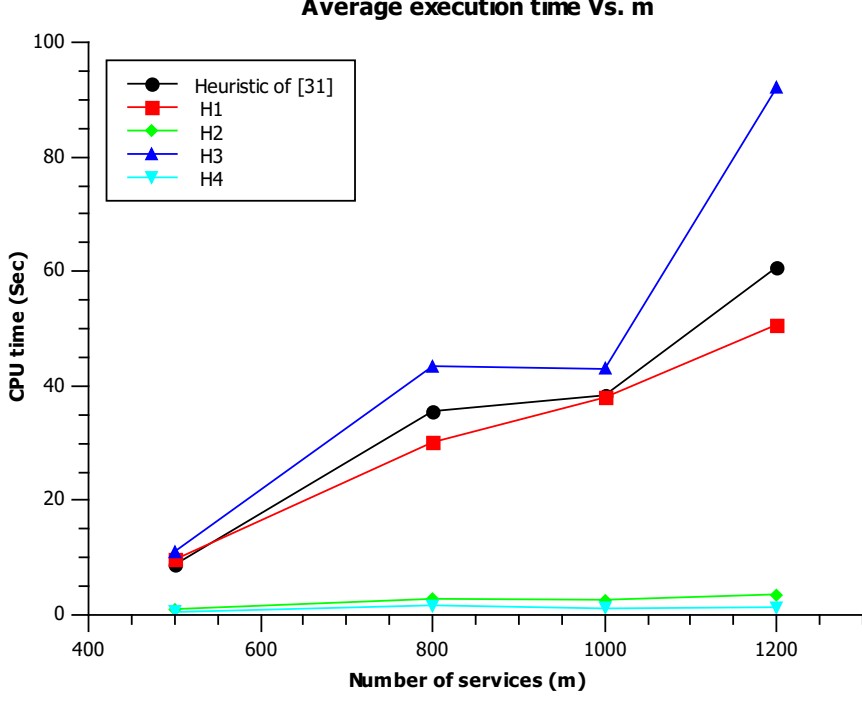

**Figure 4.** Average CPU time vs. m [31].

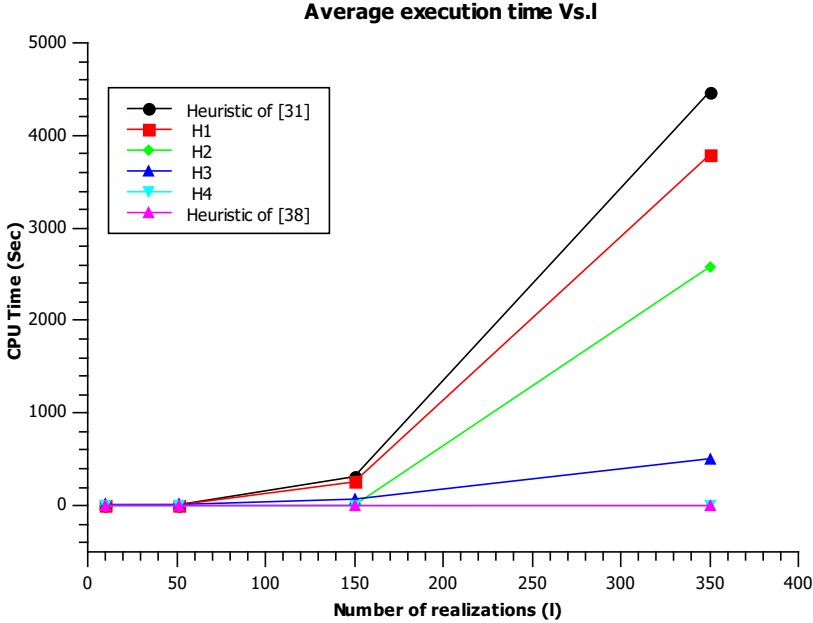

**Figure 5.** Average CPU time vs. l [31,38].

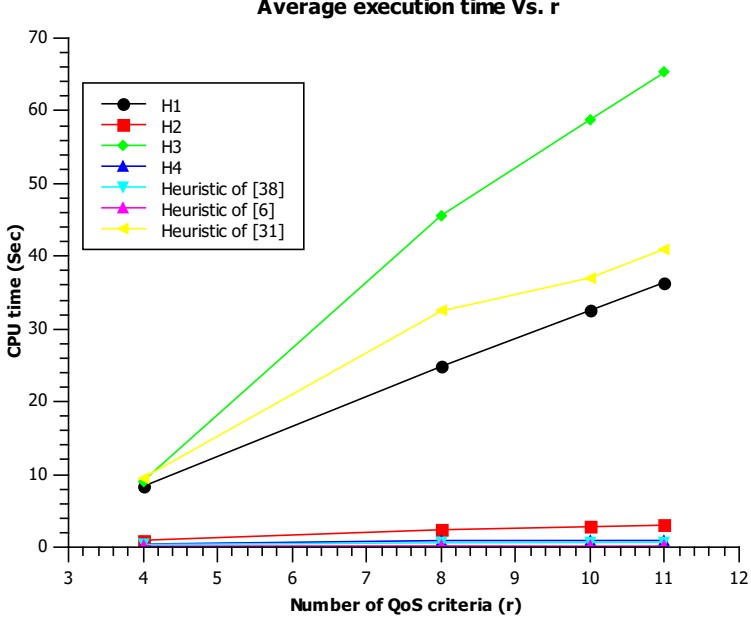

**Figure 6.** Average CPU time vs. r [6,31,38].

According to Figure 7, we observed that all methods had almost the same CPU time up to n = 5. Beyond this threshold, the time rose with different scales (according to each alternative). We noticed that the exhaustive search was the most-prohibitive one since there was an exponential number of candidate solutions; however, the DBA (with all configurations) only explores a polynomial number of candidate compositions (but the GQC is still exponential). As a result, the increasing rate of time was less drastic for the three configurations of the DBA. In summary, we can state that a selection problem with less than eight tasks can be efficiently handled by the DBA while using less than 100 bats. It is worth noting that the majority of real-world workflows have less than 10 abstract tasks, and this fact highlights the suitability of the DBA for the QoS-aware service-selection problem.

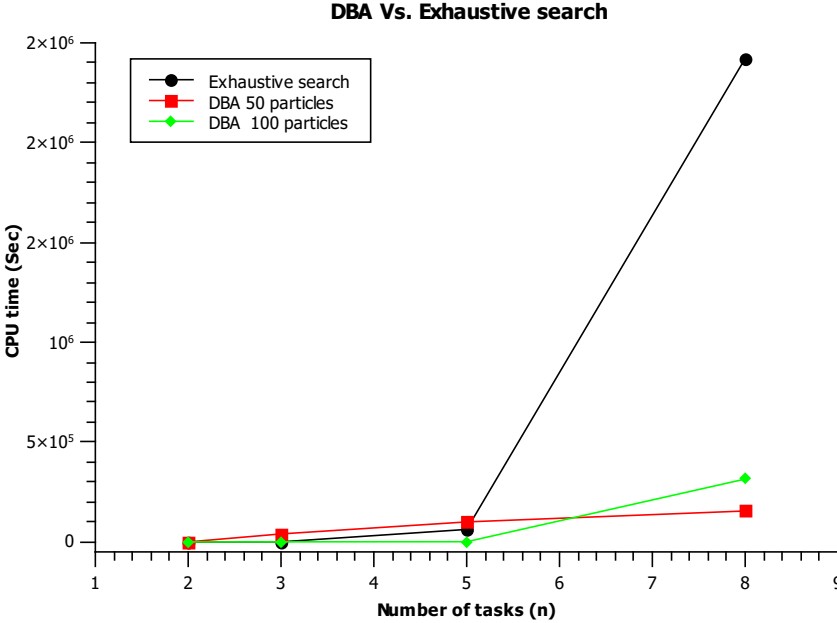

**Figure 7.** Average CPU time for DBA and exhaustive search.

Table 5 demonstrates the behavior of the heuristics with respect to the number of QoS attributes r. We noticed a general deterioration of GQC and PSGC (for all heuristics) when r grew. This deterioration can be expected since GQC is a product of r probabilities that are related to the r attributes. Moreover, we observe that H2 is the most performing heuristic for all values of r and for all methods.

**Table 5.** GQC and global constraint satisfiability vs. r.

| Model | r = 4 | | r = 8 | | r = 10 | |
|---|---|---|---|---|---|---|
| | GQC | PSGC | GQC | PSGC | GQC | PSGC |
| $H_1$ | **0.673** | **75%** | **0.484** | **50%** | **0.500** | **40%** |
| | 0.646 | 75% | 0.480 | 62.5% | 0.480 | 20% |
| $H_2$ | **0.721** | **100%** | **0.515** | **37.5%** | **0.570** | **30%** |
| | 0.664 | 100% | 0.515 | 37.5% | 0.463 | 30% |
| $H_3$ | 0.302 | 0% | 0.393 | 0% | 0.388 | 0% |
| | 0.253 | 0% | 0.343 | 0% | 0.356 | 0% |
| $H_4$ | 0.562 | 50% | 0.6628 | 12.5% | 0.408 | 10% |
| | 0.486 | 50% | 0.524 | 25% | 0.388 | 20% |
| Fuzzy dominance heuristic of [31] | 0.673 | 75% | 0.5155 | 37.5% | 0.500 | 50% |
| | 0.633 | 50% | 0.515 | 50% | 0.441 | 10% |

Table 6 shows the behavior of the heuristics with respect to the QoS sample size l. Broadly speaking, we noticed that both the GQC and PSGC degraded as l grew. This degradation is logical since the satisfaction of tight global constraints will be rare as l increases. We observed that H1 and H2 were more effective than the remaining heuristics; more specifically, H2 performed better than H1 for low values of l (we can even obtain a 100% PSGC); however, H1 performed better for medium and large values of l. In contrast to the heuristics H2, H3, and H4, we observed that H1 had a stable and consistent performance for all values of l.

**Table 6.** GQC and global constraint satisfiability vs. l.

| | l = 15 | | l = 21 | | l = 100 | |
|---|---|---|---|---|---|---|
| **Model** | **GQC** | **PSGC** | **GQC** | **PSGC** | **GQC** | **PSGC** |
| $H_1$ | **0.673** | **75%** | **0.655** | **100%** | **0.420** | **0%** |
| | **0.646** | **75%** | **0.544** | **50%** | **0.414** | **0%** |
| $H_2$ | **0.721** | **100%** | **0.704** | **50%** | **0.408** | **0%** |
| | **0.664** | **100%** | **0.655** | **100%** | **0.402** | **0%** |
| $H_3$ | 0.302 | 0% | 0.343 | 0% | 0.346 | 0% |
| | 0.253 | 0% | 0.311 | 0% | 0.324 | 0% |
| $H_4$ | 0.562 | 50% | 0.538 | 25% | 0.392 | 0% |
| | 0.486 | 50% | 0.467 | 25% | 0.390 | 0% |
| Fuzzy dominance heuristic of [31] | 0.673 | 75% | 0.665 | 75% | 0.415 | 0% |
| | 0.633 | 50% | 0.588 | 75% | 0.411 | 0% |

Table 7 presents the performance of the heuristics with respect to m (the cardinal of the task). We observed a slight degradation for both the GQC and PSGC when the number of services m increased (for almost all heuristics). This observation may be due to the fact that the new extended dataset has less-promising QoS levels. We also noticed that the heuristics H1 and H2 were more effective than the rest of the alternatives (for all values of m).

**Table 7.** GQC and global constraint satisfiability vs. m.

| | m = 500 | | m = 800 | | m = 1000 | |
|---|---|---|---|---|---|---|
| **Model** | **GQC** | **PSGC** | **GQC** | **PSGC** | **GQC** | **PSGC** |
| $H_1$ | **0.673** | **75%** | **0.645** | **75%** | **0.549** | **50%** |
| | **0.646** | **75%** | **0.626** | **75%** | **0.491** | **25%** |
| $H_2$ | **0.721** | **100%** | **0.604** | **50%** | **0.552** | **50%** |
| | **0.664** | **100%** | **0.583** | **75%** | **0.486** | **75%** |
| $H_3$ | 0.302 | 0% | 0.358 | 0% | 0.379 | 0% |
| | 0.253 | 0% | 0.311 | 0% | 0.299 | 0% |
| $H_4$ | 0.562 | 50% | 0.638 | 50% | 0.506 | 25% |
| | 0.486 | 50% | 0.620 | 25% | 0.474 | 25% |
| Fuzzy dominance heuristic of [31] | 0.673 | 75% | 0.590 | 75% | 0.551 | 50% |
| | 0.633 | 50% | 0.583 | 75% | 0.551 | 50% |

Table 8 demonstrates the performance of the heuristics with respect to the number of tasks n. It is clearly shown that the scores given by all heuristics degraded with the increasing of n, since it is more difficult to satisfy a constraint comprised of a larger sum of random variables (according to the central limit theorem, this sum will follow—under some conditions—a Gaussian probability distribution with a narrower standard deviation). Like the precedent experiments, we noticed that H1 performed better than the rest of the heuristics for all values of n. Additionally, we note that H2 had a better GQC and PSGC for low values of n, but these scores drastically degraded when n increased.

**Table 8.** GQC and global constraint satisfiability vs. n.

| | n = 2 | | n = 5 | | n = 8 | |
|---|---|---|---|---|---|---|
| Model | GQC | PSGC | GQC | PSGC | GQC | PSGC |
| $H_1$ | **0.673** | **75%** | **0.665** | **50%** | **0.609** | **50%** |
| | **0.646** | **75%** | **0.656** | **50%** | **0.592** | **75%** |
| $H_2$ | **0.721** | **100%** | **0.647** | **75%** | **0.224** | **0%** |
| | **0.664** | **100%** | **0.640** | **50%** | **0.219** | **0%** |
| $H_3$ | 0.302 | 0% | 0.163 | 0% | 0.132 | 0% |
| | 0.253 | 0% | 0.161 | 0% | 0.126 | 0% |
| $H_4$ | 0.562 | 50% | 0.557 | 25% | 0.512 | 25% |
| | 0.486 | 50% | 0.541 | 25% | 0.511 | 25% |
| Fuzzy dominance heuristic of [31] | 0.673 | 75% | 0.563 | 50% | 0.590 | 50% |
| | 0.633 | 50% | 0.557 | 50% | 0.580 | 50% |

Table 9 presents a comparison between our contributions (the DBA with H1 and H2) and some existing state-of-the-art approaches. It is clearly shown that the GQC and PSGC of H1 and H2 were more effective than the works of the literature. We also observed that the work of [6] gave the lowest values for the GQC, and this means that the methods based on local threshold selection have low performances on practical datasets. We also observed that the fuzzy implementation of the Pareto dominance using [7] was better than that of [31], since the experiments shown in Tables 5–8 confirmed the slight superiority of our proposed formula.

**Table 9.** Utility score, global constraint satisfiability, and GQC for all methods (default configuration).

| Heuristic | GQC | US | PSGC |
|---|---|---|---|
| $H_1$ | **0.655** | **0.531** | **75%** |
| | **0.544** | **0.482** | **100%** |
| $H_2$ | **0.704** | **0.511** | **50%** |
| | **0.655** | **0.531** | **100%** |
| $H_3$ | 0.342 | 0.387 | 0% |
| | 0.311 | 0.374 | 0% |
| $H_4$ | 0.538 | 0.451 | 25% |
| | 0.467 | 0.427 | 25% |
| Majority grade with constraint programming [38] | 0.703 | 0.519 | 75% |
| | 0.631 | 0.527 | 75% |
| Fuzzy dominance heuristic of [31] | 0.665 | 0.516 | 75% |
| | 0.588 | 0.514 | 75% |
| First assignment of [6] | 0.302 | 0.398 | 0% |

## 6. Conclusions

We presented in this paper a set of ranking heuristics coupled with a bat algorithm metaheuristic for selecting service compositions with an uncertain QoS. The main idea of the proposition consists of lowering the space size by first retaining the most-pertinent services in each class (task) using well-defined heuristics. In the second phase, we performed a global search to obtain the best compositions in terms of global QOS conformance. The results confirmed the ability of both the fuzzy Pareto dominance relationship and stochastic dominance (of order zero) to outperform the remaining heuristics.

In future works, we plan to test the framework on other types of workflows and compare our bat algorithm method with recent metaheuristics such as spider monkey optimization and the whale optimization algorithm. Moreover, we will also consider other alternatives for modeling uncertainty such as intuitionistic fuzzy logic and possibilistic logic.

**Author Contributions:** Conceptualization: A.E., F.H. and A.B.; data curation: A.E.; formal analysis: A.E. and F.H.; investigation: M.F. and M.K.; methodology: M.F. and S.K.; project administration: F.H.; software: A.E.; supervision: F.H. and A.B.; validation: A.E., F.H. and A.B.; writing—original draft: A.E. and F.H.; writing—review and editing: A.E., F.H. and A.B. All authors have read and agreed to the published version of the manuscript.

**Funding:** This research received no external funding.

**Institutional Review Board Statement:** Not applicable.

**Informed Consent Statement:** Not applicable.

**Data Availability Statement:** The data is generated using probability distributions with well known parameters (see Table 4).

**Conflicts of Interest:** The authors declare no conflict of interest.

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
