# Peer review of "An Intelligent Bat Algorithm for Web Service Selection with QoS Uncertainty"

_2504-2289, doi:10.3390/bdcc7030140_

Round 1

Reviewer 1 Report

The overall language of the article is poor and needs to be optimized. The analysis of the introduction is insufficient and needs to be supplemented. At the same time, the structure of the article needs to be adjusted. Significant changes should be made before publication. My comments are as follows:

1. In the Introduction part of the paper, it is not recommended to use tables and examples to describe it, especially in the first paragraph, it is necessary to supplement the background information of the relevant theories. (line 28, line 43) at the same time, the introduction lacks a summary of relevant theories at home and abroad. I think this part should be supplemented.

2. Where is the Intelligent in Intelligent Bat Algorithm embodied?

3. The author needs to explain why the search space has been reduced from mn to kn, line 53.

4. In lines 57-82, I think the summary of the contribution of the article is more appropriate in the conclusion section. Ask the author to make appropriate modifications.

5. The use of proper nouns should be consistent in the context, for example, the abbreviations of SOC in the abstract are not the same as those in the text. I think the author also needs to explain the technical terms in the text, such as the meaning of words such as "TopK" and "GQC".

6. In line 96, the author states "Many works and reviews have been proposed to address this kind of issues ([1], [4], [5], [6]). How do readers know what specific work and reviews have been put forward, they need to express their views instead of using paper citations to replace the content, and most of the later citations need to be revised. Such as 104 lines, 107 lines, 110 lines, 113 lines …. in the same chapter. Also, some papers of service composition need to be referenced.

[1] Domain Constraints-Driven Automatic Service Composition for Online Land Cover Geoprocessing

[2] Multi-Agent Planning for Automatic Geospatial Web Service Composition in Geoportals

[3] A web service-oriented geoprocessing system for supporting intelligent land cover change detection

[4] Context-aware QoS prediction for web service recommendation and selection

7. In sections 2.1 and 2.2 of this paper, the author lists the current research situation at home and abroad, so what is the relationship between these contents and the research of this paper, and what is the significance of these two sections? What information can we get about these research contents? I think the author should classify and analyze these contents instead of simply enumerating them.

8. The formula in this paper requires the author to explain the meaning and function of the parameters in detail. At the same time, the author needs to explain what expected goal we can achieve through these formulas.

9. The quality of the picture in this paper needs to be strengthened, in which there needs to be a certain gap between words and words. Figure 2

10. Figure 3 the title and picture position need to be adjusted. I don't think the placement of figure 3 is very meaningful, it's just a metaphor.

11. Please check lines 380, 391, 392 for referencing errors.

12. In the explanation of the code snippet in Section 4.3, you need to pay attention to paragraph indentation.

13. The positions of tables 7 and 8 need to be adjusted.

14. The ranking heuristic algorithm and discrete bat algorithm are applied to the service composition problem. The experimental results show that the running efficiency is higher and the performance is better than the existing algorithms, but when the QoS sample size is large and the number of tasks increases, can this method achieve conditional constraints better?

15. What are the shortcomings of the method proposed by the author?

The overall language of the article is poor and needs to be optimized. 

Reviewer 2 Report

1. Please recheck it for possible mistakes and typos. For example, "the size the available services" should be changed to "the size of the available services" in the Introduction.

2. Add a table to summarize the review of the works presented in the introduction and state-of-the-art sections.

3. Figure 1 should be redrawn. Why there is a line to task n. If it is sequential, the parallel line is not allowable.

4. All formulas extracted from previous references should be distinguished. It means that these formulas need to be referred to the references.

5. The main optimization problem should be defined as a problem with one objective function and some constraints.

6. Why alpha and Gamma are selected 0.8 and 0.8 in Algorithm 1.

7. Before section 5, complexity order is presented. It must be rewritten. You can factorize Popsize in four terms.

8. The title (caption) for figures should be below the figure. Please change their locations.

9. Figures 4, 5, 6, and 7 should be redrawn because only three cases are examined. Do it for more points.

10. All methods cited in tables and figures should be marked by [ ]. Please change ( ) to [ ] in the mentioned items.

Please check it carefully for possible mistakes to typos.

Reviewer 3 Report

The paper studies the problem of selecting web services with uncertain QoS. The authors propose a two-stage approach that reduces the search space using heuristics for ranking the tasks’ services and a bat algorithm metaheuristic for selecting the final near optimal compositions.

In the Introduction, the main notions used in the paper such as cloud computing, Quality of Service (QoS), Global QoS Conformance (GQC) , etc. are mentioned briefly. Some of the notions must be explained in more details. In particular, the important notion of QoS. I recommend to the authors to include references to the documents International Telecommunication Union for example the QoS regulations (ITU-T Supp. 9 of E.800 Series),  the vocabulary for performance, quality of service and quality of experience, etc.

Section 2, represents a literature overview on the topic. It is extensive and up-to-date but some very important works are missing. The authors have correctly addressed the fuzzy approach and more specifically the fuzzy dominance function but they have not included recent studies on QoS under uncertainty in service compositions. There are recent works which propose three intuitionistic fuzzy estimations of uncertainty in service compositions which can be applied to problem discussed in the present paper. Furthemore, an important work on QoS-aware service composition is the paper:

Strunk, A. QoS-aware service composition: A survey. In Proceedings of the 2010 Eighth IEEE European Conference on Web Services, 1–3 December 2010; pp. 67–74.

Another important recent paper from an MDPI journal that must be included deals with intuitionistic fuzzy estimations of uncertainty as a QoS indicator is

Poryazov, S. et al. Two Approaches to the Traffic Quality Intuitionistic Fuzzy Estimation of Service Compositions. Mathematics 2022, 10, 4439. https://doi.org/10.3390/math10234439.

In it, a  novel approach to the QoS estimation of service compositions based on the notion of intuitionistic fuzzy sets is proposed. The importance of the proposed approach is due to the fact that it can be applied to all types of service systems. Also, the intuitionistic fuzzy sets based approach is a more advanced one because it is based on a notion (intuitionistic fuzzy set) which is an extension of the fuzzy sets of Lotfi Zadeh.

The authors should also include references to papers on QoS estimation in certain types of service compositions such as parallel composition, distributive composition, concomitant composition etc., and the problems which appear in them such as the need of overall model normalization towards adequate presentation and estimation of QoS (and QoE).

Another critical remark: it is not accepted to start a sentence with a literature source as the authors have done multiple times in subsection 2.1.

The service selection architecture is presented very well and the mathematical model is correct.

Figure titles should be placed under the figure (see Fig. 3). The title of Fig. 3 is not clearly visible as it overlaps with figure.

The experimental results seem to be correctly obtained and they verify the model.

The Conclusion section is too short for such a comprehensive study and should be extended. I recommend to the authors to include as a future direction of research the intuitionistic fuzzy approach to the modelling of uncertainty in the models of QoS estimation.

Overall, the paper represents some important contributions to service selection under uncertain QoS. I recommend that the paper be published once the authors address adequately my remarks.

English language is fine. Minor spell check is required. There are some style issues. For example, throughout subsection 2.1 it is not accepted for a sentence to begin with a reference.

Round 2

Reviewer 1 Report

I think it is ready for publicaition.

Reviewer 2 Report

All comments are addressed carefully.

It is good.

Reviewer 3 Report

Thank you for taking into account my remarks. I recommend that the paper be published in the present form.

Minor spellcheck is required.